# Should Prenatal Chromosomal Microarray Analysis Be Offered for Pulmonary Atresia? A Single-Center Retrospective Study in China

**DOI:** 10.3390/genes14030722

**Published:** 2023-03-15

**Authors:** You Wang, Chunling Ma, Fang Fu, Hang Zhou, Ken Cheng, Ruibin Huang, Ru Li, Dongzhi Li, Can Liao

**Affiliations:** 1The First School of Clinical Medicine, Southern Medical University, Guangzhou 510515, China; 2Department of Prenatal Diagnostic Center, Guangzhou Women and Children’s Medical Center, Guangzhou Medical University, Guangzhou 510620, China; 3School of Medicine, South China University of Technology, Guangzhou 510641, China

**Keywords:** pulmonary atresia, chromosomal microarray, copy number variants, prenatal diagnosis, perinatal outcome

## Abstract

(1) Objective: To evaluate the application of chromosomal microarray analysis (CMA) in fetuses with pulmonary atresia (PA) and to explore the risk factors for predicting chromosomal imbalances and adverse perinatal outcomes. (2) Methods: This study investigated 428 cases of PA singleton pregnancies that were tested using CMA and quantitative fluorescent polymerase chain reaction (QF-PCR) as first-line genetic testing. The PA cases were divided into two groups: an isolated group and a non-isolated group. (3) Results: CMA revealed clinically relevant copy number variations (CNVs) in 9/139 (6.47%) PA fetuses, i.e., pathogenic copy number variations (pCNVs) in 8/139 (5.76%) fetuses and likely pathogenic CNVs in 1/139 (0.72%) fetuses. Stratified analysis showed that the incidence of clinically significant variants was higher in non-isolated PA fetuses than in isolated PA fetuses (12.50%, 6/48 vs. 3.30%, 3/91, *p* = 0.036). Regression analysis showed that a combination of other structural abnormalities at diagnosis of PA represented the principal risk factor for chromosomal imbalances (OR = 2.672). A combination of other structural abnormalities and a high maternal age increased the risk of adverse pregnancy outcomes in PA cases, including intrauterine fetal death (IUFD), termination of pregnancy (TOP), and preterm delivery. (4) Conclusions: The value of CMA for locating imbalanced genetic variations in fetuses with PA was highlighted by this study, particularly when combined with additional structural abnormalities.

## 1. Introduction

Congenital heart defects (CHDs) are among the most common congenital malformations worldwide, having an impact on 3–11 out of every 1000 live births and being the main cause of newborn death [1]. CHDs are the leading global cause of newborn morbidity and mortality and place a heavy economic burden on the world. Although the etiology of most types of CHD is still not fully known, various genetic and environmental variables have been proven to be involved in the disease during the past few decades. CHDs have been shown to be associated with a wide variety of complicated chromosomal abnormalities, submicroscopic rearrangements, whole-chromosome aneuploidies, and single-gene alterations. Almost eighty percent of those with 22q11.2 deletion syndrome are born with cardiovascular malformations. Tetralogy of Fallot, tetralogy of Fallot with pulmonary atresia, and pulmonary atresia with the ventricular septal defect are the most common defects among them. Pulmonary atresia (PA) is a severe cyanotic congenital heart disease in which there is no direct connection between the right ventricle and the pulmonary artery. The development of the pulmonary vasculature influences surgical interventions, whether they be palliative or drastic. The distribution of the pulmonary vasculature and the collateral artery flow determine the severity of the cyanosis. PA accounts for 1% to 3% of congenital heart diseases [2,3]. According to the integrity of the ventricular septum, PA is classified as pulmonary atresia with an intact ventricular septum (PA/IVS) or pulmonary atresia with a ventricular septal defect (PA/VSD). Despite sharing a pulmonary atresia family cousinship, the morphological anomalies, hemodynamics, surgical approach, and prognosis seem to be very different. With advances in the genetic understanding of congenital heart defects, it has become evident that chromosomal or genetic variants underlie a significant fraction of CHDs [4]. Due to hypoxic tissue ischemia before surgery and extracorporeal circulation during surgery, infants with PA are susceptible to postoperative neurological complications [5]. Infants with genetic conditions are prone to cognitive impairment and postoperative complications; their mortality rate is significantly higher than that of infants without genetic conditions. The multisystemic involvement (e.g., impaired swallowing function, immunodeficiency, hypocalcemia) might complicate postoperative management [5,6].

A chromosomal microarray (CMA) is a technique that detects chromosomal imbalances known as copy number variations (CNVs) [7,8,9]. The application of CMA to ultrasonic structural defects has been extensively studied and recommended. In fetuses with ultrasound structural anomalies, the detection rate of pathogenic copy number variations (pCNVs) is approximately 6.0% [10,11,12,13,14,15], and clinical and research-based testing suggests that CNVs contribute to 10–15% of CHDs [16,17]. The detection rate obtained by CMA for CHD is slightly higher than that for other structural abnormalities. In recent years, many studies have applied CMA in the genetic diagnosis of congenital heart disease, and CMA has been recommended as the first-line clinical diagnostic tool [15,17].

In this study, we used genome-wide high-resolution CMA technology to perform genomic analyses on PA fetuses to investigate the genetic etiology of PA and the application value of CMA technology in PA fetal cases.

## 2. Materials and Methods

### 2.1. Research Participants

This was a retrospective cohort study of 146 fetuses with or without other abnormalities that were diagnosed with PA by prenatal ultrasound or fetal echocardiography at the Guangzhou Women and Children’s Medical Center from January 2017 to January 2022. The study protocol was approved by the Guangzhou Women and Children’s Medical Center’s ethics committee. All patients agreed that their data could be used in the study. The survey was carried out in accordance with relevant guidelines and regulations. Inclusion criteria were (I) PA diagnosed by echocardiography; (II) accurate gestational ages based on the mother’s menstrual history and the results of her first-trimester scan; (III) all enrolled cases subjected to invasive prenatal genetic testing. Exclusion criteria were (I) multiple pregnancies; (II) maternal age (MA) < 18 years old; (III) suspected TORCH infection (toxoplasma, cytomegalovirus, rubella virus, herpes simplex virus, and others).

Fetal echocardiography, using grayscale and color Doppler ultrasound, was performed by ultrasound experts to assess the fetal heart. Each patient underwent standard echocardiography for 30–40 min to review the fetal cardiac anatomy and to record any abnormalities. If there was no direct connection between the right fetal ventricle and the pulmonary artery, fetal PA was diagnosed according to this ultrasound scan. PA fetuses were divided into two groups based on whether they were combined with other structural abnormalities: an isolated group, and a non-isolated group.

An electronic ultrasound database and patient records were used to gather information about maternal and fetal clinical characteristics and perinatal outcomes. These included maternal age (MA), pregnancy history, gestational age (GA) at PA diagnosis, gestational age at invasive prenatal diagnosis, karyotype and CMA results, the outcome of pregnancy, GA at birth, mode of delivery, physical examination of newborns, and postnatal therapy if necessary. Patients were routinely contacted by phone for follow-up interviews to collect information. Clinical postnatal follow-up evaluations were scheduled from birth to one year.

### 2.2. Chromosome Microarray Analysis

#### 2.2.1. CMA Detection

Fetal DNA was extracted from amniocytes and umbilical blood samples using QIAamp DNA Blood Mini Kit (Qiagen GmbH, Hilden, Germany) to screen for aneuploidy on chromosomes 13, 18, 21, X, and Y and to rule out maternal cell contamination by quantitative fluorescence polymerase chain reaction (QF-PCR) [18]. If the QF-PCR results showed aneuploidy, the CMA was canceled. Karyotype analysis was carried out via the traditional G-band approach (550-band resolution). According to the manufacturer’s instructions, CMA was carried out using an Affymetrix CytoScan HD/750K array with a single-nucleotide polymorphism array (SNP array) and array-based comparative genomic hybridization (aCGH) platforms at resolutions of 10 and 100 kb, respectively (Affymetrix Inc., Santa Clara, CA, United States). ChAS software was used for data analysis. The Cytoscan HD chip contains 750,000 SNP probes and 195,000 oligonucleotide probes covering the entire genome. It can detect abnormalities in chromosome numbers, deletion/duplication of large segments, and CNVs (<5 MB), as well as loss of heterozygosity (LOH), uniparental diploidy (UPD), and mosaicism (mosaicism ratio > 10%).

#### 2.2.2. CMA Results Analysis

CNVs ≥ 100 kb in size were selected for comparative analysis. CNVs are categorized into five categories in accordance with the joint consensus guidelines of the American College of Medical Genetics (ACMG) and Clinical Genome Resource (ClinGen): pathogenic, likely pathogenic, variations of unknown significance (VOUS), likely benign, and benign. Following GRCh37/hg19 genome construction, genomic locations were assessed. With the use of the International System for Human Cytogenomic Nomenclature, the CMA and karyotype findings were recorded (ISCN 2020). In addition, as part of the CMA results for heritability evaluation or trios’ analysis, DNA collected from mother and paternal blood samples was also analyzed. VOUS, likely pathogenic CNVs, and pathogenic CNVs were recorded, but likely benign variants were not considered. Two authors (R.L. and F.F.) reviewed all reported CNVs to ensure that the classifications were fully updated based on the knowledge and experience described in public databases. The public databases interrogated in the process of variant classification included ClinVar (https://www.ncbi.nlm.nih.gov/clinvar/, accessed on 23 January 2020), OMIM (https://www.omim.org, accessed on 23 January 2020), DECIPHER (http://decipher.sanger.ac.uk/, accessed on 23 February 2020), ClinGen resource (https://www.clinicalgenome.org/, accessed on 24 February 2020), University of California Santa Cruz (UCSC, http://genome.ucsc.edu/hg19, accessed on 24 February 2020), and ISCA (https://www.Iscaconsortium.org/, accessed on 25 February 2020).

#### 2.2.3. CMA Results Validation

All CNVs were further confirmed by fluorescence in situ hybridization (FISH) or real-time polymerase chain reaction (RT-PCR). Verification was performed not only on fetal samples but also on parental samples.

### 2.3. Clinical Follow-Up

Pregnancy outcomes included live birth, termination of pregnancy (TOP), or neonatal death. We conducted clinical follow-up assessments by means of electronic medical records or telephone enquiry one year after birth, and routine follow-up assessments were performed annually. If other cardiac structural abnormalities are found during the screening of newborns, we recommended that the physicians at the Heart Center perform a further evaluation.

### 2.4. Statistical Analysis

Statistical analysis was performed using Pearson’s chi-square or Fisher’s exact test and SPSS software version 25.0 (IBM); a *p* value < 0.05 was considered statistically significant.

## 3. Results

PA was identified in 148 singleton pregnancies in total, and the prenatal diagnoses were performed by invasive methods at Guangzhou Women and Children’s Medical Center from January 2017 to January 2022. Table 1 lists the characteristics of the pregnant women and fetuses in the study cohort. Four (2.74%) of these 148 PA fetuses were lost to follow-up during the period of PA’s natural history. Two cases had TORCH-positive results. In addition, QF-PCR detected three cases of chromosomal abnormalities, i.e., one case of trisomy 21, one case of trisomy 18, and one case of sex chromosome number anomaly (47, XYY) confirmed by chromosomal karyotype analysis. Finally, 139 PA cases with complete natural history data were analyzed in detail.

Amniocentesis for genetic diagnosis was performed in 101 (72.66%) cases, and percutaneous umbilical cord blood collection was performed in the remaining 38 cases (27.34%). Chromosomal microarrays discovered clinically significant variants in 9/139 (6.47%) fetuses, including pathogenic copy number variations in 8/139 (5.76%) fetuses and likely pathogenic CNVs in 1/139 (0.72%) fetuses. VOUS were discovered in 18/139 (12.95%) fetuses. Parental CMA results were available in seven cases with P/LP or VOUS, but the remaining patients refused the test due to economic reasons. The results of prenatal diagnosis and the details of follow-up are recorded in the flowchart in Figure 1.

The mean maternal age (MA) (at the time of reporting) was 30.1 ± 3.8 years old, and the median gestational age (GA) at diagnosis of PA was 23^+2^ (21^+5^–27^+4^) weeks. In all instances, 103/139 (74.10%) were diagnosed in the late second trimester (19^+6^–27^+5^ weeks of pregnancy), and the remaining 36/139 (25.90%) were diagnosed in the third trimester. Among all cases, 85 (61.15%) were live births, 52 (37.41%) chose to have the pregnancy terminated, and a diagnosis of spontaneous intrauterine fetal death was made in 2 (1.44%) cases. Table 2 shows the clinical and chromosome characteristics of nine PA cases with clinically significant variation (pathogenic CNVs (*n* = 8) and likely pathogenic CNVs (*n* = 1)). Of these nine cases with clinically significant variation, eight had CNV fragments <10 Mb, and only one had CNV fragments >10 Mb. Of the nine clinically significant CNVs, eight were copy-number losses, and one was a duplication. The molecular diagnoses were 22q11.2 microdeletion syndrome (*n* = 4), Williams–Beuren syndrome (*n* = 1), mosaic chromosome 21 monosomy (*n* = 1), 8q24.3 duplication with 21q22.2q22.3 deletion (*n* = 1), 16p13.11 microdeletion syndrome (*n* = 1), and 7q36.1 microdeletion (*n* = 1).

Table 3 provides a statistical analysis of the genetic and clinical outcomes of the PA fetuses. Of the 85 live births, 67 (73.62%) cases were isolated PA. A total of 56 cases were normally delivered; six cases were delivered prematurely, and 23 were by cesarean section. Three cases of atrial septal defects and one case of bilateral clubfoot that were missed by prenatal ultrasound were found during newborn screening. Prenatal ultrasound discovered seven cases of VSD and five cases of tetralogy of Fallot. These cases were confirmed after birth, and seven individuals had successful postpartum surgical therapy. Fifty-six newborns were confirmed to have PA and underwent surgery after birth. While four fetuses had a bad prognosis due to growth and psychomotor retardation, the others were surgically treated and are now doing well. A total of seven (77.78%) of the nine pregnant women with positive genetic test results decided to terminate their pregnancy after learning that the CMA test results were abnormal. Information obtained from the two surviving infants showed that one was found to have unilateral clubfoot after birth, but no abnormalities were found in the other infant.

Regression analysis showed that a combination of other structural abnormalities at diagnosis of PA (OR = 2.672, *p* = 0.019) was the main risk factor for the prediction of chromosomal imbalances in PA fetuses. Figure 2 shows the results of logistic regression analysis for predicting the risk factors of adverse pregnancy outcomes in PA cases.

## 4. Discussion

PA is a complex cardiac cone deformity defined as the lack of a continuous lumen between the ventricle and the pulmonary artery. Its incidence rate is 0.42/10,000 live births, and PA can be associated with structural anomalies affecting other systems [1]. Its hemodynamic characteristics are that the right ventricle flows into the left cardiac system through ASD or VSD, and the pulmonary artery blood flow originates entirely from the aortic system. Without surgery, the early mortality rate can be as high as 80%; however, the operation process is extremely complex, and this represents the main difficulty of current treatment.

Our research leads us to conclude that genetic conditions are among of the high-risk factors affecting fetal PA survival prognosis, but this is inconsistent with other research findings. For example, Michielon et al. proposed that 22q11.2 microdeletion syndrome and trisomy 21 syndrome had no effect on fetal PA prognosis [19]. The blockage of the heart and main vessels’ embryonic development may be a factor in PA. [20]. Both the development of the right ventricular outflow tract and the formation and degeneration of the fourth and sixth pairs of pharyngeal arch arteries are related to the formation of PA [21]. Some animal experiments have also shown that PA formation is related to the development of the second heart area [20].

PA is a right-heart obstructive heart disease that relies on the arterial duct for pulmonary blood flow; the fetus depends on the unclosed arterial duct to maintain the blood supply to the lungs after birth. Suppose the collateral circulation between extracorporeal circulation and pulmonary circulation is good; in that case, even if the hypoxic condition is mild, it can cause heart failure, and some children even die in the neonatal period due to heart failure. Therefore, some scholars have developed a clinical prognostic scoring system for fetal cardiac structural malformations, emphasizing adopting different delivery-delivery room models and multidisciplinary treatment strategies based on PA anatomical classification and disease severity. With the in-depth study of the pathology and hemodynamics of pulmonary atresia and the continuous progress of surgical techniques, the surgical treatment effect of PA children has been continuously improved. Therefore, timely and accurate prenatal diagnosis and classification of PA has important guiding significance for clinical eugenic selection and timely postpartum treatment.

With this study, we identified the prevalence and distribution of genetic variations in PA fetuses, whether or not they had other structural defects. We also explored and described the clinical outcomes of the PA fetuses, as well as the factors affecting prognosis. Most importantly, our study confirmed the value of CMA in detecting genomic imbalances in PA fetuses, particularly fetuses suspected of having PA in combination with other structural abnormalities. Overall, CMA had a clinically significant variant detection rate of 6.47% in PA fetuses. However, the detection rate of clinically important variants in the non-isolated PA group was higher than that in the isolated group (12.50% vs. 3.30%, *p* < 0.05). The logistic analysis showed that a combination with other malformations represented the main risk factor for chromosomal imbalances when diagnosing PA. In combination with other structural malformations, maternal age of >35 years was a positive predictor of abnormal CMA results in PA fetuses. In summary, MA, GA in diagnosing PA, and the presence of other structural abnormalities all increase the risk of adverse pregnancy outcomes in PA fetuses. In providing patients with prenatally identified PA counseling, these objective data may be useful.

Considering our results relating to chromosomal imbalances, we found that 22q11.2 microdeletions are particularly common in PA fetuses. Among the nine pathogenic CNVs detected in our cohort, four (44.44%) involved 22q11.2 microdeletion syndrome. In addition to this, 22q11.2 microdeletion syndrome; trisomy 21, 18, and 13; and Turner’s syndrome are chromosomal imbalances commonly found in clinical practice and are common chromosomal imbalances in CHDs. Furthermore, 22q11.2 microdeletion syndrome is the most common microdeletion syndrome, although its gene-fragment deletion size is about 1.5–3.0 Mbp [22]. Additionally, 22q11 microdeletion syndrome mainly manifests as cardiac malformation, a cleft palate, hypocalcemia, thymic dysplasia, facial abnormalities, etc. The deleted key gene (*TBX1*) may be involved in developing the sixth arched artery; it may also be involved in the junction between the peripheral and central pulmonary arteries. Notably, in 80% of cases, the causative imbalance of 22q11.2 microdeletion syndrome is not inherited and is defined as de novo. However, regardless of their sex, affected individuals, in common with patients with an autosomal dominant condition, face a 50% chance of having an affected child in each pregnancy. For this reason, 22q11.2 microdeletion syndrome is a focus of current research [22].

Turner syndrome, 22q11.2 microdeletion syndrome, 21-trisomy syndrome, 18-trisomy syndrome, and 13-trisomy syndrome are common chromosomal defects in the clinic and are also common chromosomal defects in congenital heart disease. In 1998, Hofbeck et al. [23] used the D22S75 probe to detect 22q11.2 microdeletion in 21 cases of PA combined with VSD and MAPCAs. The results showed that in 10 cases of 22q11.2 microdeletion-positive patients, except for one case involving a single supply of pulmonary blood and the complete branch-like structure of the bilateral pulmonary arteries, the rest of the positive patients had multiple pulmonary blood sources and no complete pulmonary artery branches. In contrast, eight cases of negatively tested children all had a single source of pulmonary blood and complete bilateral pulmonary artery branches. This showed that 22q11.2 microdeletion syndrome is associated with the occurrence of PA and is particularly strongly correlated with PA from multiple pulmonary blood sources. These results were confirmed in our study.

It is worth noting that PA/VSD has a more significant correlation with 22q11 microdeletion than PA/IVS, especially in fetuses with MAPCAs and/or right aortic arch or thymus dysplasia, which increases the risk of 22q11 microdeletion. Therefore, fetal craniofacial and thymic development should be observed when PA/VSD is detected on prenatal ultrasound of the fetal heart. In this study, both fetuses showed thymus dysplasia, facial morphological abnormalities, and PA/VSD, and their genetic testing results showed 22q11 microdeletion. To sum up, when prenatal ultrasound finds that the fetus has the above developmental abnormalities, pregnant women should be recommended to undergo genetic testing to determine whether the PA fetus is combined with 22q11 microdeletion.

In about 40% of reported cases, trisomy 21 is associated with CHDs, such as complete atrioventricular canal defect (CAVC), VSD, ASD, and TOF; however, PA is rarely reported [24]. Using a model of trisomy 21 syndromes in mice, Roper et al. [25] confirmed that affected animals had lower numbers of neural crest cells and significantly weaker ability to migrate and proliferate compared with diploid mice, and the expression of Sonic Hedgehog (SHH) was also significantly weakened. SHH knockout mice exhibited hypoplasia of the ventral pharyngeal arch tail, their fourth and sixth pairs of pharyngeal arch arteries were poorly developed, and they exhibited PA [25]. This shows that trisomy 21 plays a certain role in the pathogenesis of PA. The single positive case in our study confirmed this. Unfortunately, in this case, after the parents knew the karyotype results, they decided to terminate the pregnancy, so it could not be further determined whether a 22q11.2 microdeletion was present.

In recent years, with the development of high-resolution ultrasound instruments and the improvement of the technical level of sonographers, the application of transabdominal ultrasound, transvaginal ultrasound, or a combination of the two can better display the fetal heart structure in early and late pregnancy. Several studies have shown that the combination of fetal nuchal translucency (NT) and ductus venosus blood flow spectrum abnormalities can improve the specificity and accuracy of screening for congenital heart defects [26,27,28,29]. In this study, four PA fetuses showed NT thickening in the first trimester of pregnancy, and two exhibited a-wave reversal in the ductus venosus. Any prenatal ultrasound examination in early pregnancy that reveals fetal NT thickening and/or ductus venosus blood flow spectrum abnormalities is of potentially great value, and it is also necessary to determine whether the fetus has PA. In this regard, it is worth mentioning that fetuses without pulmonary artery forward flow need to be distinguished from fetuses with pulmonary artery pseudo atresia caused by severe pulmonary hypertension. A fetal period due to various reasons (such as diaphragmatic hernia) leads to increased pulmonary artery pressure or resistance, tricuspid regurgitation causes much pulmonary artery forward flow to be reduced or even disappear, and severe right ventricular dysfunction and twin transfusion syndrome recipients can also be manifested as false pulmonary atresia. Pseudo-pulmonary atresia usually causes right heart enlargement, while pulmonary atresia often causes right heart dysplasia, which is an important difference between the two. Furthermore, the prenatal detection of PA in early pregnancy enables early clinical diagnosis and decision-making, alleviating the pain and psychological trauma of pregnant women who choose to terminate the pregnancy at a later gestational age and improving perinatal survival.

It is essential to carry out postnatal follow-ups of children with a prenatal diagnosis of fetal PA. To this end, we collected data on the pregnancy outcomes and postpartum treatment of the live-birth PA cases described above. A total of 59 (69.4%) of the 85 live births in our study cohort underwent postpartum surgery and had a good prognosis. Our data revealed that non-isolated PA is a powerful predictor of adverse outcomes in cases of PA pregnancy, including termination of pregnancy (TOP), intrauterine fetal death (IUFD), and preterm birth. We also found that cases of isolated PA have a better prenatal and postnatal prognosis than cases of non-isolated PA and that fetal PA associated with other anomalies correlates with increased rates of pregnancy termination and postnatal surgery. Furthermore, our statistical analysis showed that the prognosis of PA fetuses without genetic conditions was better than that of PA fetuses with genetic conditions (64.63% vs. 11.11%, *p* < 0.05). Overall, a definitive genetic diagnosis and a good prognosis help to increase the confidence of patients to continue a pregnancy, especially those who carry fetuses with isolated PA and have negative genetic testing results.

We acknowledge that our research has several limitations. Firstly, this study is retrospective and has inherent limitations, requiring larger prospective studies. Secondly, our data were obtained from a single center and involved patients of similar ethnic backgrounds, so our study may be considered to lack heterogeneity. Thirdly, our means of detection may have overlooked balanced chromosomal rearrangements. Fourthly, due to the failure to obtain familial consent, we were unable to obtain available postmortem analysis results. Finally, the long-term prognosis is still unknown because the oldest patient was aged only one year at follow-up.

## 5. Conclusions

To our knowledge, this is the first largest prenatal study using CMA for fetal PA chromosome detection. This study emphasized the importance of chromosome detection for PA fetuses and found that non-isolated PA fetuses had a higher rate of chromosomal imbalances. In this study, we found that cases of isolated PA had better prenatal and postpartum prognosis compared with non-isolated PA cases (73.62% vs. 37.50%, *p* = 0.001) and that non-isolated PA led to an increase in the pregnancy termination rate and the postpartum surgery rate. Our study data revealed that CMA produced a genetic etiology diagnosis rate of 6.47% for prenatal PA and that the detection rate in non-isolated PA cases was higher than that of the isolated group (12.50%, 6/48 vs. 3.30%, 3/91, *p* < 0.05). Finally, we found that CNVs were identified in a substantial proportion of PA cases. In conclusion, we recommend that CMA should be offered in all pregnancies with fetal PA, regardless of whether other ultrasound anomalies are present or absent.

## Figures and Tables

**Figure 1 genes-14-00722-f001:**
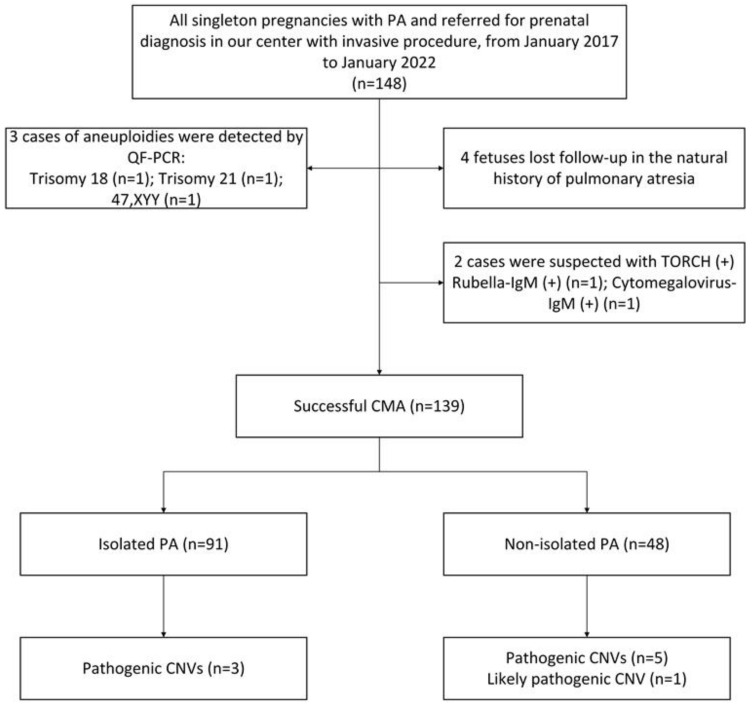
Flow chart of the study cohort.

**Figure 2 genes-14-00722-f002:**
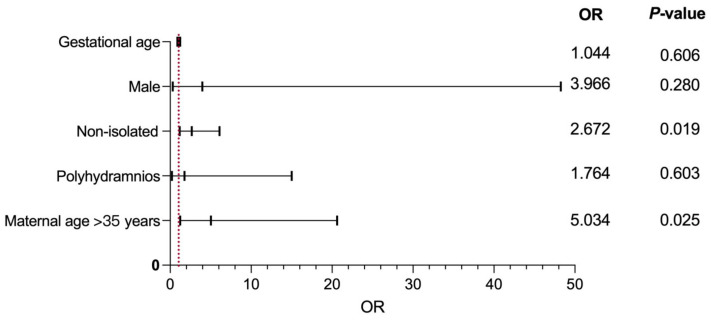
Binary logistic regression analysis of factors leading to the predictors of adverse pregnancy outcomes of PA. OR, odds ratio.

**Table 1 genes-14-00722-t001:** Characteristics of the research population’s mothers and fetuses.

Maternal Age (Median)	30.1 (Range 19.3–47.2) Years
Gestational weeks (median) ^A^	23.3 (range 21.7–27.6) weeks
Sex of fetuses (M/F)	68/71
Sample types	
Amniotic fluid	101
Cord blood	38
Malformation classification	
Isolated	98
Non-isolated	41
Follow-ups ^B^	139

^A^ Means the gestational age at CA diagnosis. ^B^ Successful follow-up cases.

**Table 2 genes-14-00722-t002:** Pathogenic and likely pathogenic copy number variations identified in the cohort of fetuses with pulmonary atresia.

Patient	MA	GA at the Suspicion of PA	Associated Anomaly	CMA Results	Length	Type	Classification	Outcome	Parental Study
1	33.1	32.4	kidney agenesis, hemivertebrae	arr[hg19] 22q11.21(18916842–21465659) × 1	2.55 Mb	Deletion	P	spontaneous labor	de novo
2	34.25	27.1	single ventricle	arr[hg19] 22q11.21(18648856–21800471) × 1	3.15 Mb	Deletion	P	termination of pregnancy	de novo
3	28.2	21.71	hemivertebrae	arr[hg19] 22q11.21(18648856–21800471) × 1	3.15 Mb	Deletion	P	cesarean section	de novo
4	29.07	26.5	VSD	arr[hg19] 22q11.21(18631365–21800471) × 1	3.17 Mb	Deletion	P	termination of pregnancy	de novo
5	25.76	28.2	hemivertebrae	arr[hg19] 7q11.23(72557180–74628840) × 1	2.07 Mb	Deletion	P	termination of pregnancy	de novo
6	19.31	33.21	double outlet right ventricle, VSD	arr[hg19] 21q11.2q22.3(15016487–48093361) × 1~2 mos	33.08 Mb	Mosaic deletion	P	termination of pregnancy	de novo
7	27.59	29.71	VSD	arr[hg19] 8q24.3(140131302–146295771) × 3 arr[hg19] 21q22.2q22.3(39737188–48093361) × 1	6.16 Mb 8.36 Mb	Duplication Deletion	P P	termination of pregnancy	Paternally inherited
8	47.31	29.7	VSD	arr[hg19] 16p13.11(15481921–16388343) × 1	906 Kb	Deletion	P	termination of pregnancy	de novo
9	25.67	27.63	VSD	arr[hg19] 7q36.1(152092064–152341146) × 1	249 Kb	Deletion	LP	termination of pregnancy	de novo

MA: Maternal age; GA, gestational age; VSD, ventricular septal defect; P: Pathogenic; LP: Likely pathogenic.

**Table 3 genes-14-00722-t003:** Statistical analysis of the genetic results and perinatal outcomes of PA fetuses.

Perinatal Outcome	Total (n = 139)	Isolated PA (n = 91)	Non-Isolated PA (n = 48)	*p*-Value	CMA Testing Results	Isolated PA (n = 91)	Non-Isolated PA (n = 48)	*p*-Value
TOP	52 (37.41%)	23 (14.28%)	29 (60.42%)	0.000	CNVs (n = 9)	3 (3.30%)	6 (12.50%)	0.036
Live birth	85 (61.15%)	67 (73.62%)	18 (37.50%)	0.001	VOUS (n = 18)	12 (13.19%)	6 (12.50%)	0.909
Intrauterine death	2 (1.44%)	1 (1.10%)	1 (2.08%)	0.643	negative (n = 112)	76 (83.52%)	36 (75.00%)	0.228

PA: pulmonary atresia; CNVs: clinically significant variants; VOUS: variants of unknown significance; TOP: termination of pregnancy.

## Data Availability

The original contributions presented in the study are included in the article; further inquiries can be directed to the corresponding author.

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
