# Peer review of "Should Prenatal Chromosomal Microarray Analysis Be Offered for Pulmonary Atresia? A Single-Center Retrospective Study in China"

_genes, 2023, doi:10.3390/genes14030722_

Round 1
Reviewer 1 Report
Dear authors,
The data presented deserve publishing, but English grammar and syntax result in obscure sentences and in a very tedious review process. This is disrespectful towards the journal and reviewers. Please provide a significant revision of English Language.
Page 1
Abstract
Line 17
“Pulmonary atresia (PA) fetuses”
Replace with “Fetuses with Pulmonary Atresia (PA)”
Lines 17-18
“chromosomal aberrations”
The term chromosomal aberration should not be used anymore. “Chromosomal imbalances” is a valuable synonym.
Line 20
“gene detection strategy”
The wording is inappropriate. “Genetic testing” might be a suitable substitute.
Lines 21-22
“Comparison of chromosome abnormality rate by PA subgroup (isolated and non-isolated).”
This sentence has no verb, nor meaning. Please revise.
Lines 26-28
“Regression analysis showed that the PA fetus, combined with other structural abnormalities, was the main risk factor for chromosome aberration (OR=2.672)”
The meaning is unclear.
Lines 30-32
“This 30 study highlights the value of CMA in the imbalance of PA fetal chromosome subgroups, especially PA with other structural abnormalities.”
The meaning is unclear. The term “imbalance” is inappropriate in this position. Please revise
Introduction
Line 37
“The Congenital heart defects”
Remove the article “The”. If you introduce the CHD acronym, the words should be spelled with the first letter in uppercase (Congenital Heart Defetcs).
Lines 38-39
“About cases per 1000 live births are affected, which is the leading cause of neonatal deaths“
Poor English language. Please revise
Pages 1 and 2
Lines 44-45
“It has been found that many congenital heart defects have genetic defects”
Poor wording. Replace with “it has become evident that chromosomal or genetic variants underly a significant fraction of CHDs”.
Page2
Line 47
“genetic defects”
Replace with “genetic conditions”. Same for line 49, 51 and other parts o the text.
Lines 49-51
“[…], and the complication of other systemic diseases (e.g., impaired swallowing function, immunodeficiency, hypocalcemia) complicates postoperative management”
Poor English language. Edit as : “[…]. The multisystemic involvement (e.g., impaired swallowing function, immunodeficiency, hypocalcemia) might complicate postoperative management.”
Lines 52-53
“[…] has important guiding significance for clinical eugenic selection”
This statement is unethical. Please, remove or revise.
Line 54
“is a newly developed technique”
CMA is not recent compared to many other technologies. Please remove the statement.
Lines 56-58
“In fetuses with ultrasound structural anomalies, the detection rate of pathogenic copy number variations (pCNVs) is approximately 6.0%”
Please provide more updated references (meta-analysis if available or systematic reveiews) on the diagnostic yield of CMA in fetuses with structural anomalies and, if possible, compare them with the results of analogous references exploring the yield of CMA in cardiovascular malformations
Lines 58-60
“In recent years, many studies have applied CMA 58 to the genetic diagnosis of congenital heart disease and have recommended CMA as the 59 first-line clinical diagnostic tool.”
Add a reference at the end of the sentence.
Materials and Methods
Line 67
“diagnosed as PA”
Replace with “diagnosed with PA”
Line 73
“should be required to access gestational age”
Remove
Line 75
“infection”
The term is repeated before and after the brackets (…). Please, remove one.
Lines 80-81
“If there is no direct connection between the right fetal ventricle and the pulmonary artery, fetal PA is diagnosed according to this ultrasound scan”
Please replace the present tense “is” with the past tense “was”
Line 86
“Invasive surgery”
Do you mean invasive prenatal diagnosis?
Lines 91-95
Up to this point standard G-banding Karyotipe was not introduced. Also, if the section 2.2 is named “chromosome microarray analysis”, the name “routine chromosome karyotype analysis” is not appropriate for a sub-section. Pleawse, edit
Page
Lines 98-99
“Invasive samples were analyzed with quantitative fluorescent polymerase chain reaction (QF-PCR) by utilizing a 99 multiplex ligation-dependent probe amplification (MLPA) kit ”
It is not clear how you use a MLPA kit to perform a Qf-PCR assay. Please, disclose the method or provide a citation.
Line 113
“mosaic”
Replace with “mosaicism”
Line 115
“fragments”
Replace with “in size”
Line 122
“typically”
What do you mean by using this adverb in this case? If there were exceptions to this method, it should be disclosed.
Line 126
“A few open databases have been used to classify CNVs including”
Reword as “The public databases interrogated in the process of variant classification included”
Line 134
“QF-PCR verified the detected pathogenic CNVs”
Did you perform QF-PCR validation assays only on pathogenic variants? Were these assays performed on fetal samples only or also on parents? Did any of the cases require FISH confirmation?
Page4
Results
Line 147
“invasive surgery”
See above
Line 150
“2 cases of TORH positive results”
It is not advised to start a sentence with an Arabic number (2). Replace with the word “Two2. Also, the sentence lacks the verb. Please, edit
Line 152
“chromosome number abnormality”
Edit as “sex chromosome number anomaly”
Line 159
“VOUS was found”
Edit as “VOUS were found”
Line 161
“because it cost more than USD 1000”
Replace with “due to economical reasons”
Lines 172-173
“In these 9 cases of CNVs, except for 1 case with duplication, all other cases have missing CNVs”
Replace with “Of the 9 clinically significant CNVs, 8 were copy-number losses while 1 was a duplication2
The sentence is repeated a second time in lines 173-174. Please remove.
Lines 174-177
“The phenotypes involved […] microdeletion (n=1)”
You can reword as “The molecular diagnoses were”
Lines 182-185
As stated in the comment on the abstract, the meaning of these sentences is obscure and they should be extensively reworked.
Page 5
Figure 1
The font size is too small and the image resolution is low.
Fugure 2
Please add a more comprehensive explanation in the caption of the figure.
Page 6
Table 2
Reword the title as “ Pathogenic and Likely Pathogenic Copy Number Variations identified in the cohort of fetuses with Pulmonary Atresia”.
In the table, reword “induced abortion” as “termination of pregnancy”
Table 3
Pleas add a caption explaining the table results
Page 7
Discussion
Line 202
“and almost all extracardiac malformations can be combined”
Reword as “ and can be associated with structural anomalies affecting other systems.
Line 205-206
“but the surgery[…] treatment”
Avoid repetitions.
Lines 206-208
“The genetic defect […] affect prognosis”
Poor English language. Please, reword. Also, as stated ina previous sections, do not use the term “genetic defect”
Line 211
“were formed”
Replace with “are formed”
Line 216
“unexpected”
Poor word choice. Please, revise.
Lines 219-211
See comments above concerning the ethical concerns on the expression “clinical eugenic selection”. Also, there is no correlation between this conclusion and the paragraph.
Line 236
“genetic aberration”
See comment in previous section
Line 251
“gene fragment deletion syndrome”
The definition is not appropriate.
Line 253
“the deleted gene”
Are you referring do TBX1? If yes please state it clearly in the text.
Lines 254-255
“Notably […] de novo”
A clinical phenotype, when not inherited, is defined as “sporadic”. A genetic variant, if not inherited, is defined as “de novo”. Please edit the text.
Lines 258-261
Obscure meaning. Please, revise extensively
Line 263
“About 40 % of trisomy”
Add “of cases” between “40%” and “of trisomy”.
Lines 262-319
This whole section of the Discussion is wordy and poorly written. An extensive English language revision is required.
Page 9
Conclusions
The whole section requires extensive English language revision. In addition, CMA is already a first-tier test in fetuses with cardiovascular anomalies. The conclusion should focus more on the definition of a Pa-specific CMA diagnostic yield and on data on pregnancy and postnatal outcomes, which can be useful for genetic counseling.
Author Response
Dear Editor and Reviewer 1,
Thank you very much for giving us an opportunity to revise our manuscript entitled, " Should prenatal chromosomal microarray analysis be offered for pulmonary atresia? A single-center retrospective study from China" Genes-2256512. We appreciate the time and effort that you and the reviewers dedicated to providing feedback on our manuscript and are grateful for the insightful comments on and valuable improvements to our paper. We have carefully studied the reviewer's comments carefully and tried our best to revise according to the comments. The language editing is carried out by an English expert (MDPI English Language Editing Services). The English editing certificate has been uploaded (ID: english-61680). Revised portions are marked in red in the revised paper.
We have incorporated most of the suggestions made by the reviewers. Those changes are highlighted within the manuscript. Please see below for a point-by-point response to the reviewer's comments and concerns. All page numbers refer to the revised manuscript file with tracked changes.
Thank you very much for your attention and consideration.
We would like also to thank you for allowing us to resubmit a revised copy of the manuscript.
We hope that the revised manuscript is accepted for publication in Genes.

Reviewer 2 Report
I thank the editor’s invitation to review the manuscript entitled “Should prenatal chromosomal microarray analysis be offered for pulmonary atresia? A single-center retrospective study from China”, sent for publication in Genes. This manuscript describes a retrospective study analyzing characteristics of 428 cases of pulmonary atresia identified prenatally in fetuses in which a smaller group of fetuses (n=139) were evaluated in characteristics of chromosome microarray analysis performed during diagnostic work-up. Significant CNVs (definitely and likely pathogenic variants) were observed in less than 7% of cases, and in most cases with pathogenic changes identified by genetic testing pulmonary atresia was only part of a more complex structural disease, including complex malformities syndromes. It is of note that 7 of the 9 cases with significant CNVs identified after genetic testing evolved with induced abortion. The manuscript brings important contributions both for current literature and clinical practice, and authors should take a look at some points:
1. In Table 3, there is a minor typo: “VOUS”, which should be described as VUS.
2. Taking into account the risks involved in the procedures to collect amniocytes and/or umbilical blood for genetic analysis, do the authors think that chromosomal microarray should be routinely performed prenatally to evaluate fetuses with suspected complex congenital cardiopathy or pulmonary atresia (in syndromic and malformation contexts)? Would it be feasible to evaluate other sources for genetic evaluation (in a non-invasive approach), considering that few specific therapeutic purposes would be done in cases with definite genetic changes identified? Despite proper genetic counseling during the prenatal period, what would be the advantages of performing such evaluation at the prenatal stage?
3. Are results of postmortem analysis of fetuses available? Maybe these results will enable the identification of clinical aspects still unknown for some of the chromosomal abnormalities and deletions and duplications identified in the sample.
4. In the “Funding” topic, there is probably a minor distraction regarding a lack of substitution of the term “Please add”, which should be eliminated.
Author Response
Dear Editor and Reviewer 2,
Thank you very much for giving us an opportunity to revise our manuscript entitled, " Should prenatal chromosomal microarray analysis be offered for pulmonary atresia? A single-center retrospective study from China" Genes-2256512. We appreciate the time and effort that you and the reviewers dedicated to providing feedback on our manuscript and are grateful for the insightful comments on and valuable improvements to our paper. We have carefully studied the reviewer's comments carefully and tried our best to revise according to the comments. The language editing is carried out by an English expert.
We have incorporated most of the suggestions made by the reviewers. Those changes are highlighted within the manuscript. Please see below for a point-by-point response to the reviewer's comments and concerns. All page numbers refer to the revised manuscript file with tracked changes.
Thank you very much for your attention and consideration.
We would like also to thank you for allowing us to resubmit a revised copy of the manuscript.
We hope that the revised manuscript is accepted for publication in Genes.

Round 2
Reviewer 1 Report
Thank you for revising the English grammar and syntax of the paper.
This has emended most of the issues of the original draft.
I do however believe that some minor improvements are required.
Lines 95-99
“Invasive samples were analyzed 95 with quantitative fluorescent polymerase chain reaction (QF-PCR) using a multiplex ligation-dependent probe amplification (MLPA) kit to screen for aneuploidy on chromosomes 13, 18, 21, X, and Y or to exclude maternal cell contamination (Guangzhou Darui Biotech-98 nology Co., Ltd, Guangdong, China)”
The added reference does not address techniques for rapid aneuploidy detection. Please, address my concerns about the differences between QF-PCR and MLPA kits expressed in the previous review rounds
Lines 179-180
“Information obtained for these nine infants showed that one was 179 found to have unilateral clubfoot after birth, but no abnormalities were found in the other 180 infants.”
If seven of the nine pregnancies mentioned ended in TOP, there would only be two infants, and not nine as declared. Please, revise.
Table 3
What do the authors mean by “induced abortion”? What is the difference with “termination of pregnancy”?
Lines 249-250
“80% of the genetic variation of 22q11.2 microdeletion syndrome is not inherited, and it is defined as “de novo”.”
Reword as” in 80% of cases the causative imbalance of 22q11.2 microdeletion syndrome is not inherited, and is defined as de novo “
Line 276
“venous catheter wave reversal.”
I am not sure this is the correct nomenclature, especially concerning the term “catheter”. (also in line 280)
ADDITIONAL REMARKS
The initial section of the discussion, focusing on the embryology of pulmonary atresia and the role of common chromosomal anomalies such as trisomy 21 an 22q11.2 del should should be significantly shorter as, albeit interesting, it is only partially relevant to the results of the study.
Author Response
Dear Editor and Reviewer 1,
Thank you very much for giving us another chance to revise our manuscript entitled, " Should prenatal chromosomal microarray analysis be offered for pulmonary atresia? A single-center retrospective study from China" Genes-2256512. We appreciate the time and effort that you and the reviewers dedicated to providing feedback on our manuscript and are grateful for the insightful comments on and valuable improvements to our paper. We have carefully studied the reviewer's comments carefully and tried our best to revise according to the comments. The language editing is carried out by an English expert (MDPI English Language Editing Services). The English editing certificate has been uploaded (ID: english-61680). Revised portions are marked in red in the revised paper.
We have incorporated most of the suggestions made by the reviewers. Those changes are highlighted within the manuscript. Please see below for a point-by-point response to the reviewer's comments and concerns. All page numbers refer to the revised manuscript file with tracked changes.
Thank you very much for your attention and consideration.
We would like also to thank you for allowing us to resubmit a revised copy of the manuscript.
We hope that the revised manuscript is accepted for publication in Genes.
